



".

# Using Optimization Tools to Explore Stratospheric Aerosol Injection Strategies

Ezra Brody[1,*], Yan Zhang[1,*], Douglas G. MacMartin[1], Daniele Visioni[2], Ben Kravitz[3,6], and Ewa M. Bednarz[1,4,5]

[1]Sibley School of Mechanical and Aerospace Engineering, Cornell University, Ithaca, NY, USA
[2]Department of Earth and Atmospheric Sciences, Cornell University, Ithaca, NY, USA
[3]Earth and Atmospheric Sciences, Indiana University, Bloomington, IN, USA
[4]Cooperative Institute for Research in Environmental Sciences (CIRES), University of Colorado Boulder, Boulder, CO, USA
[5]NOAA Chemical Sciences Laboratory, Boulder, CO, USA
[6]Atmospheric Sciences and Global Change Division, Pacific Northwest National Laboratory, Richland, WA, USA
[*]These authors contributed equally to this work.

**Correspondence:** Ezra Brody (eb637@cornell.edu)

**Abstract.** Stratospheric aerosol injection (SAI), as a possible supplement to emission reduction, has the potential to reduce some of the impacts associated with climate change. However, the outcomes will depend on how it is deployed: not just how much, but the latitudes of injection and the distribution of injection rates across those latitudes. Different such strategies have

5 been proposed, managing up to three climate metrics simultaneously by injecting at multiple latitudes. Nonetheless, these strategies still do not fully compensate for the pattern of climate changes caused by increased greenhouse gas concentrations, creating a novel climate state. To date there has not been a systematic assessment of whether there are strategies that could do a better job of managing some specific climate goals, nor an assessment of any underlying trade-offs between managing different sets of climate goals. Herein we use existing climate model simulations of the response to injection at 7 different latitudes, and

10 apply optimization tools to explore the limitations and trade-offs when designing strategies that combine injection across these latitudes. This relies on linearity being a sufficiently good assumption, which we first validate. The resulting "best" strategy of course depends on what goals are being optimized for. For example, at 1 degree Celsius of cooling, we predict that there exist strategies that do a better job than those simulated to date at simultaneously balancing regional temperature and precipitation responses, but the differences may be too small to detect at lower levels of cooling.

## 1 Introduction

Stratospheric aerosol injection (SAI) is being studied as a possible supplement to emission reduction to temporarily mitigate some of the risks associated with climate change. This approach would reduce global mean temperature by adding aerosols, or their precursors, to the lower stratosphere, to increase solar reflection before it reaches the surface. Existing research has shown that SAI does not bring the climate back to a previous state, but creates a novel climate (Bala et al., 2010; Irvine et al.,

2010). The choice of how much to inject at different latitude(s) and seasons(s) we refer to as the strategy, and it is clear that



different strategies will lead to different climate outcomes (Robock et al., 2008; Ban-Weiss and Caldeira, 2010; MacMartin et al., 2013; Kravitz et al., 2016, 2019; Lee et al., 2020, 2023; Bednarz et al., 2023a, b; Zhang et al., 2024; Wells et al., 2024). As such, one cannot evaluate the benefits and risks associated with SAI without first considering the choice of strategy. To do this, it is important to better understand the design space: for any given set of climate goals, does there exist a strategy that can simultaneously meet them, or what is the "best" strategy to balance those goals? How much does the strategy itself then depend on the chosen goals? How much do the outcomes then depend on what the strategy is designed for – that is, are there underlying trade-offs between achieving different goals, and how much does this matter for assessing the outcomes?

Most existing studies evaluate the potential climate impact of SAI by simulating some particular defined strategies in global climate models. Few studies attempt to explore the broader range of possible climate responses (Ban-Weiss and Caldeira, 2010; MacMartin et al., 2013; Lee et al., 2020). Ban-Weiss and Caldeira (2010) first treated SAI as an optimization problem, with three degrees of freedom of specified aerosols optimized for several climate goals. MacMartin et al. (2013) explore the trade-offs between different climate goals, but based on idealized patterns of solar reduction, via reducing the incoming solar radiation directly in the climate model in multiple different spatial and seasonal patterns. This study takes a similar approach in principle to these pioneering studies, but for the first time evaluates trade-offs between multiple different climate goals based on simulations of stratospheric aerosol injection, albeit in a single climate model.

A key enabler for this research is understanding how "big" the design space for SAI is, that is, how many different latitudes or seasons of injection can be chosen to optimize over before the differences from adding additional degrees of freedom are too small to matter: injection at two different latitudes very close to each other might technically have a different influence on the climate system, but the differences would not be detectable over a reasonable time-frame. The design space here is chosen based on the conclusion from Zhang et al. (2022), that there are of order 6-8 degrees of freedom available for a cooling of up to 1-1.5°C (though it is important to recognize that this is quite a considerable amount of cooling and not a level that would likely be reached for several decades after the start of any deployment, if ever). We consider annually-constant injection at the equator, 15°N and S, 30°N and S, and spring-time injection at 60°N and S, described more in the next sections; these reasonably span the space.

Herein we use existing climate model simulations for injection at these latitudes or combinations of latitudes (MacMartin et al., 2022; Bednarz et al., 2022, 2023b, a; Zhang et al., 2024), and use optimization tools to determine the "best" SAI strategy – that is, the best combination of injection rates across the 7 latitudes above – to minimize different objective functions. This allows us to address the questions above. If a climate model simulation were conducted with the injection rates determined from this optimization process, it of course would not exactly match the predicted outcomes, and thus it is critical to contextualize the differences between outcomes when optimizing for different metrics relative to both the error introduced by the assumption of linearity, and the uncertainty caused by natural variability. The latter influences both how confident we are in how the climate model might respond to a particular choice of injection rates, and also how well an observer could distinguish between different strategies – are the differences large enough to matter? The answer to that clearly depends on how much warming is being offset by SAI as the forced response to SAI will scale with the cooling while the variability will not.





The next section describes the climate model and the existing simulations that we use in our analyses. Section 3 then describes extracting a linear model from simulation output, the validation that linearity is a reasonable approximation, and the formulation as a constrained optimization problem. Section 4 explores different trade-offs. First, if the number of climate goals is equal to or fewer than the number of degrees of freedom, then it may be possible (depending on specific goals) to find strategies that simultaneously satisfy all of the goals; a three-degree-of-freedom example of this is the multi-objective strategy

described in Kravitz et al. (2017) and subsequently used in various studies such as GLENS (Tilmes et al., 2018) and ARISE-SAI (Richter et al., 2022). Second is the more general case where there are many distinct climate goals, such as minimizing the spatial root-mean-square (rms) across the planet of the difference in annual-mean temperature or precipitation relative to some earlier reference climate of the same global mean temperature. In this case, no strategy will simultaneously satisfy all of the objectives. Changing the relative weighting between goals, for example between r.m.s. temperature versus r.m.s. precipitation

anomalies, yields a Pareto front against which existing strategies can be compared.

## 2    Climate model and Simulations

All simulations of SAI strategies considered in this study are conducted in version 2 of the Community Earth System Model with the Middle Atmosphere (MA) chemistry configuration of the Whole Atmosphere Community Climate model, version 6, as the atmospheric component, CESM2(WACCM6-MA) (Danabasoglu et al., 2020; Gettelman et al., 2019; Davis et al.,

2022). CESM2(WACCM6) is a fully coupled Earth system model consisting of atmosphere, land, ocean, sea ice, land ice, river runoff and surface waves components. The atmospheric component has 70 vertical layers that extend from the Earth's surface to about 140 km in altitude, with a horizontal resolution of 0.95° in latitude and 1.25° in longitude. Compared to the comprehensive troposphere-stratosphere-mesosphere-lower thermosphere (TSMLT) chemistry configuration, the MA version of WACCM6 used here reduces computing time by about 35% by neglecting non-methane hydrocarbon species and reactions

in the troposphere (Gettelman et al., 2019; Davis et al., 2022). While it significantly reduces the computing cost, this simplified chemistry configuration yields climate, variability, and climate sensitivity that are comparable to the TSMLT version (Davis et al., 2022).

As described earlier, Zhang et al. (2022) estimate that there are about 6-8 degrees of freedom in the SAI design space for offsetting global warming by 1-1.5 °C, and suggest that a complete set of injection choices that yield detectably different

surface climate responses includes injections at the following latitudes: 60° N, 30° N, 15° N, the equator, 15° S, 30° S, and 60° S, with all but the high-latitude cases injecting the same amount throughout the year. At high latitudes, spring injection is more efficient (Lee et al., 2021) as the aerosol lifetime is shorter and there is minimal benefit from having aerosols in the winter; injection at 60° N is thus only during March, April, and May (MAM), and injection at 60° S only in September, October, and November (SON).

A number of different climate model simulations have been conducted in CESM2(WACCM-MA) using different combinations of these injection latitudes, summarized in Table 1; all of these branch from SSP2-4.5 simulation in 2035, with continued





evolution of other non-SAI concentrations from the Shared Socioeconomic Pathway (SSP) 2-4.5, (Meinshausen et al., 2020). The SSP2-4.5 simulation is also conducted in CESM2(WACCM6-MA) from the year 2015 to 2099, with 3 ensemble members.

At each latitude, single-latitude fixed-injection-rate simulations have been conducted with 12 Tg/yr injection of $SO_2$. The setup and some results from the first 10 years of simulation are described in Visioni et al. (2023a) and Bednarz et al. (2023b) where CESM(WACCM) is compared with other climate models; these cases were extended to 35 years (from 2035 through 2069), with two ensemble members each in Bednarz et al. (2022) and a third ensemble member added in Bednarz et al. (2024, to be submitted); additional simulations (one ensemble member each) have also since been conducted for the 60°N and 60°S cases. These sensitivity experiments are intended to understand how the climate responds to a given injection location and not to represent plausible scenarios for deployment. We also use the "Arctic-high" simulation from Lee et al. (2023), where injection is only at 60°N, but with a variable injection rate to maintain Arctic September sea ice at a target level; we average the sensitivities between this case and the fixed-injection-rate simulation at the same latitude and season.

Zhang et al. (2024) simulate four hemispherically-symmetric strategies, injecting either only at the equator ("EQ"), equal amounts at 15°N and 15°S ("15N+15S"), equal amounts at 30°N and 30°S ("30N+30S"), and equal amounts in the spring at 60°N and 60°S ("60N+60S"). Each simulation uses a feedback algorithm (MacMartin et al., 2014) to adjust the injection rate to maintain the global mean temperature (denoted here as $T_0$) at 1.0°C above pre-industrial following the definition in MacMartin et al. (2022); this corresponds to maintaining temperature at the averaged value over 2008 to 2027; this time period will also be used as the reference climate below. In addition, the multi-objective strategy from Kravitz et al. (2017) has been simulated in this model with the same target for global mean temperature ; this strategy injects $SO_2$ at 15°N, 30°N, 15°S, and 30°S to maintain not just $T_0$ but also a measure of interhemispheric temperature gradient ($T_1$) and equator-to-pole temperature gradient ($T_2$) using a feedback algorithm. Maintaining $T_1$ in this climate model under SSP2-4.5 requires more injection in the southern hemisphere than the northern (Fasullo and Richter, 2023; Visioni et al., 2023b); this multi-objective strategy injects only 4% $SO_2$ at 30° N, while 15° S, 15° N, and 30° S receive about 52%, 30%, and 14% of injected $SO_2$ respectively.

Only 7 cases are actually necessary here to span the space, but the availability of additional cases allows us to assess how well a linear quasi-static approximation captures the simulated response (discussed in detail in section 3.2 below). In addition to any nonlinear dependence on injection rate, the pattern of climate response will differ slightly between a case that gradually ramps up injection rates to maintain $T_0$ and one that has a step change in injection rate, even if the final global mean temperature is similar, due in part to the two cases yielding a different state for the Atlantic Meridional Overturning Circulation (AMOC). For spanning the space here, we choose a set that both spans the design space while minimizing the errors introduced from the linear and quasi-static assumption; we thus pick to the extent possible cases that maintain constant global mean temperature rather than the fixed injection-rate cases, and cases where the change in interhemispheric temperature gradient is not too large, as we expect these situations to be those where the linear quasi-static approximation will be better. For use in the analysis in Sections 4-5 we thus use the 4 hemispherically-symmetric strategies, the multi-objective strategy, and complete that set with two single-latitude cases; 60°N and 30°N. While it would be conceptually simpler to just use the single-latitude injection cases, the ability to usefully predict the response of a linear combination of strategies is crucial here, and thus it is worth the slight extra complication to minimize as much as possible the errors from the linear quasi-static approximation. Note that





**Table 1.** Simulations of different SAI injection choices. A subset of seven of these (marked with a '*') is sufficient to span the SAI design space for a global cooling of 1-1.5°C; additional simulations are needed to assess the adequacy of the linearity approximation.

| Strategy | Injection latitude | Injection season | Injection rate | Altitude (km) | Number of ensembles |
|---|---|---|---|---|---|
| 60N* | 60° N | MAM | 12Tg/yr | 15.0 | 1 |
| 60N-SSI* | 60° N | MAM | ∼12 Tg/yr | 15.0 | 1 |
| 30N* | 30° N | year-round | 12Tg/yr | 21.5 | 3 |
| 15N | 15° N | year-round | 12Tg/yr | 21.5 | 3 |
| 0N | equator | year-round | 12Tg/yr | 21.5 | 3 |
| 15S | 15° S | year-round | 12Tg/yr | 21.5 | 3 |
| 30S | 30° S | year-round | 12Tg/yr | 21.5 | 3 |
| 60S | 60° S | SON | 12Tg/yr | 15.0 | 1 |
| 0N* | equator | year-round | variable | 21.5 | 3 |
| 15N+15S* | 15° N, 15° S | year-round | variable | 21.5 | 3 |
| 30N+30S* | 30° N, 30° S | year-round | variable | 21.5 | 3 |
| 60N+60S* | 60° N, 60° S | MAM at 60° N and SON at 60° S | variable | 15.0 | 3 |
| Multi-objective* | 30° N, 15° N, 15° S, 30° S | year-round | variable | 21.5 | 3 |

the multi-objective strategy has been used in several single-model simulation data-sets for impact analyses (e.g. GLENS and ARISE-SAI, as noted earlier), and more recently the simpler hemispherically-symmetric 30N+30S strategy has been chosen for the next set of model intercomparisons under GeoMIP (Visioni et al., 2024) , and the 60N+60S strategy has also received some attention (Goddard et al., 2023; Bednarz et al., 2022, 2023b, a). Indeed, a possible way to think about the results of optimization in Section 5 is to ask whether these existing strategies are good enough for evaluating the policy-relevant impacts of a deployment of SAI, or whether there are other strategies that should be considered more carefully.

## 3 Climate response and linearity

### 3.1 Estimating climate responses to forcings

With the deployment of SAI, the climate system is subject to both GHG and SAI forcings. The climate system's response to these external forcings is dynamic. However, for sufficiently slowly-varying forcing the pattern of climate response remains roughly constant, and scales proportionally to the changes in global mean temperature from each forcing; this is the underlying approximation in pattern-scaling that is commonly assumed in building climate emulators for example, as in Tebaldi and Arblaster (2014) and MacMartin and Kravitz (2016). An exception here is the North Atlantic where, at least in this model, the strength of AMOC responds to forcings at a much slower rate than the surface climate.

The climate system response is also nonlinear. However, provided that the response to additional external forcings relative to a reference climate state is sufficiently small, the response can be adequately approximated as linear. Again, this is a common





approximation in developing climate emulators and in past work on SAI (e.g., Moreno-Cruz et al., 2012; MacMartin et al., 2013; Farley et al., 2024). Collectively, the approximation of the climate response to forcing as being both linear and quasi-static allows us to predict how the climate model would respond to any combination of SAI injections across different latitudes, and then optimize across those injection rates for any particular choice of climate objectives. If these injection rates were then used in a simulation in the climate model, the actual response of the model would of course differ from the prediction; we will include in analysis of optimization results an estimate of how large these errors are likely to be.

Approximating the response of the climate system to forcing to be both linear and quasi-static, then the change of any climate variable $y$ due to forcing, relative to the reference climate, can be described as:

$$\Delta y = \alpha \Delta T_{0_\text{warming}} + \mu \Delta T_{0_\text{cooling}} \tag{1}$$

where $\alpha$ is the change in the climate variable $y$ in response to GHG forcing per $1°$C of global warming, $\Delta T_{0_\text{warming}}$ is the increase in global mean temperature due to increased GHG forcing, $\mu$ is the change in the climate variable in response to SAI forcing per $1°$C of global cooling, and $\Delta T_{0_\text{cooling}}$ is the decrease in global mean temperature due to SAI forcing; this is illustrated schematically in Figure 1.

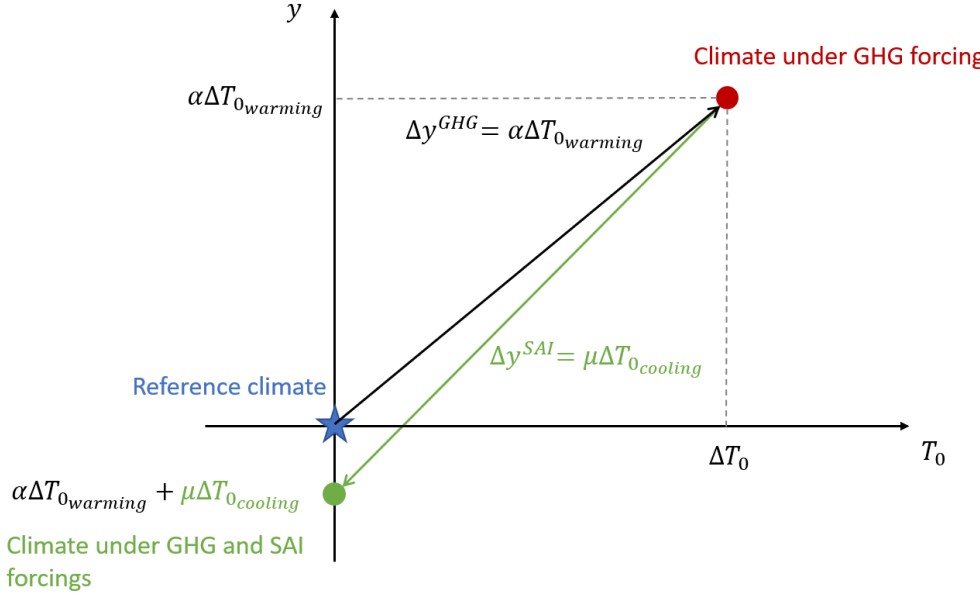

**Figure 1.** A schematic plot of the linear model of climate responses to GHG and SAI forcings, described in eq. (1), modified from Fig. 1 in Moreno-Cruz et al. (2012). The $x$-axis is the change in global mean temperature relative to the reference climate. The $y$-axis is the change in climate variable $y$ relative to the reference climate. The black line shows how the climate variable $y$ responds to GHG forcing and the green line shows how the SAI forcing modifies the climate under global warming. The $y$-axis intercept of the line with SAI corresponds to the case where the global mean temperature increase due to GHG forcing is perfectly offset by SAI; this is the case we will focus on herein.



The climate variables that we consider here include "large-scale" variables, global mean temperature ($T_0$), global mean precipitation ($P_0$), interhemispheric temperature gradient ($T_1$), equator-to-pole temperature gradient ($T_2$), intertropical convergence zone (ITCZ), and September Arctic sea ice extent (SSI), as well as surface-air temperature, precipitation, and precipitation minus evaporation (P – E) at the model grid-scale. For all of these variables we consider only annual mean changes (except SSI of course). The same methodology could be extended to consider other climate variables, including seasonal changes.

$T_1$ and $T_2$ are defined as the linear and quadratic meridional dependence of the zonal-mean temperature (Kravitz et al., 2016):

$$T_1 = \frac{1}{A} \int_{-\pi/2}^{\pi/2} T(\psi)\sin(\psi)\,\mathrm{d}A \tag{2}$$

$$T_2 = \frac{1}{A} \int_{-\pi/2}^{\pi/2} T(\psi)\frac{1}{2}(3\sin^2(\psi)-1)\,\mathrm{d}A \tag{3}$$

where $A$ is the surface area of the Earth, $\psi$ is the latitude in radians, and $T(\psi)$ is the zonal-mean temperature at latitude $\psi$.

The Intertropical Convergence Zone (ITCZ) is a band near the equator where the northeast and southeast trade winds collide (Byrne et al., 2018). Here, we define the location of ITCZ as the latitude near the equator where the zonal mean meridional streamfunction at 500 hPa changes sign. The zonal mean meridional streamfunction at each latitude is calculated based on the following equation:

$$\Psi = \frac{2\pi a \cos(\phi)}{g} \int_0^p [v]\,\mathrm{d}p' \tag{4}$$

where $a$ is the Earth's radius, $\psi$ is latitude, $[v]$ is the zonal mean meridional velocity and $p$ is 500 hPa.

To estimate the climate response to GHG forcing per $1°$C of global warming, $\alpha$, we use the existing SSP2-4.5 simulation conducted in CESM2(WACCM6-MA) (Davis et al., 2022). We estimate $\alpha$ based on the changes in climate variables due to GHG forcing over a period of time, $\Delta y^{GHG}$, and the corresponding increase in global mean temperature, $\Delta T_0$:

$$\alpha = \frac{\Delta y^{\mathrm{GHG}}}{\Delta T_{0_{\mathrm{warming}}}} \tag{5}$$

For all climate variables considered here, except SSI, we choose $\Delta y^{GHG}$ and $\Delta T_{0_{\mathrm{warming}}}$ to be the changes from the reference time period (2008–2027) to a future time period (2050–2069). For SSI, we use a closer future time period (2030–2049), because as SSI approaches zero, the response becomes more strongly nonlinear. The increase in global mean temperature and the responses of these climate variables are adequately linear over these chosen time periods.

For each case in Table 1, the climate response to different SAI forcing per $1°$C of global cooling, $\mu$, is calculated based on the changes between the reference period and the 2050–2069 time period in the CESM simulations. This difference is a result of the sum of the response to GHG forcing and the response to SAI, and thus to get the change just due to SAI we use the



estimated response to GHG forcing and rewrite equation (1) as:

$$\mu = \frac{\Delta y - \alpha \Delta T_{0_\text{warming}}}{\Delta T_{0_\text{cooling}}} \tag{6}$$

For this calculation, we use the $\alpha$ and $\Delta T_{0_\text{warming}}$ values from the SSP2-4.5 simulations, and the $\Delta y$ value from the SAI simulations listed in Table 1. The value of $\Delta T_{0_\text{cooling}}$ is calculated by subtracting $T_0$ averaged from 2050-2069 in the SAI simulations from that of the SSP2-4.5 simulations. For the seven strategies chosen to span the design space, Table 2 summarizes the values of $\alpha$ and $\mu$ for all of the large-scale climate variable discussed in this study.

**Table 2.** Climate responses to GHG forcing per $1°$C of global warming ($\alpha$) and climate responses to SAI forcing per $1°$C of global cooling ($\mu$). (The valus of $\alpha$ and $\mu$ for global mean temperature $T_0$ are both one by definition.)

| | SSP2-4.5 | 0N | 15N+15S | 30N+30S | 60N+60S | 60N | Multi-objective | 30N |
|---|---|---|---|---|---|---|---|---|
| $P_0$ [mm/day] | 0.065 | -0.087 | -0.079 | -0.072 | -0.065 | -0.057 | -0.079 | -0.064 |
| $T_1$ [°C] | 0.091 | -0.227 | -0.178 | -0.162 | -0.192 | -0.636 | -0.118 | -0.504 |
| $T_2$ [°C] | 0.072 | -0.058 | -0.082 | -0.122 | -0.223 | -0.274 | -0.084 | -0.160 |
| ITCZ [°] | -0.7 | -0.5 | -0.5 | -0.7 | -0.4 | -3.0 | 0.2 | -4.0 |
| SSI [$1\times10^6$ km$^2$] | -2.7 | 3.2 | 3.0 | 3.1 | 4.2 | 7.7 | 2.6 | 5.0 |

By assuming linearity, the change in a climate variable $y$ to a linear combination of $n$ different SAI strategies can be estimated as:

$$\Delta y = \alpha \Delta T_{0_\text{warming}} + \sum_{k=1}^{k=n} x_k \mu_k \Delta T_{0_\text{cooling}} \tag{7}$$

where $x_k$ denotes the fraction of the total global cooling provided by the $k$th strategy, $\mu_k$ denotes the corresponding change in $y$ per $1°$C of global cooling provided by the $k$th strategy, and $\Delta T_{0_\text{cooling}}$ denotes the total global cooling provided by SAI. We can enforce a constraint that SAI fully offsets the change in global mean temperature relative to the reference period by requiring $\sum_{k=1}^{k=n} x_k = 1$ (this is equivalent to selecting the reference period as the time period where the global mean temperature without SAI was the same as it is in the future world with increased GHG and with SAI). The equation above can then be simplified by defining $\beta_k = \alpha + \mu_k$ so that with $\Delta T_{0_\text{cooling}} = \Delta T_{0_\text{warming}}$,

$$\frac{\Delta y}{\Delta T_0} = \beta = \sum_{k=1}^{k=7} x_k \beta_k \tag{8}$$

### 3.2 Evaluation of the linear approximation

To evaluate the error introduced in making a linear approximation of the effective climate response to a combination of different SAI forcings, we compare the simulated response to the multi-objective strategy with a linear combination of the climate responses from other simulations that add up to the same injection rates. The multi-objective and hemispherically symmetric





simulations maintain the same constant global mean surface temperature, whereas the single-latitude simulations inject at a constant rate. In order to minimize the error introduced by this difference, we match the injection rates as much as possible using the 15N+15S and 30N+30S simulations. The multi-objective case injects 14%, 52%, 30% and 4% of the total at 30°N, 15°N, 15°S and 30°S, respectively. To match this, we take 18% of the 30N+30S case, +5% of the 30N case and −5% of the 30S case, and we take 82% of the 15N+15S case, +11% of the 15N case and −11% of the 15S case. A negative value here signifies that some percentage of the inverse of the pattern from a simulation is used.

Figure 2 shows the predicted regional response of temperature and precipitation compared to the actual simulated response. For both temperature and precipitation, the difference between the predicted and actual response is within 2 standard errors of reference-period natural variability for almost the entire planet. The spatial root-mean-square (rms) difference between the predicted and actual response is $0.088°$C for surface temperature and 0.097 mm/day for precipitation. The predicted and simulated response for large-scale variables are shown in Table 3; most are within the standard error of natural variability with the exception of the interhemispheric temperature gradient $T_1$.

**Table 3.** The predicted and simulated $\mu$ values of $T_0$, $T_1$, $T_2$, $P_0$, ITCZ, and SSI in response to multi-objective strategy.

| $\mu$ | $T_1$ [°C] | $T_2$ [°C] | $P_0$ [mm/day] | ITCZ [°] | SSI[$1 \times 10^6$ km$^2$] |
|---|---|---|---|---|---|
| Simulated value | -0.118 | -0.084 | -0.079 | 0.24 | 2.6 |
| Predicted value | -0.091 | -0.076 | -0.080 | 0.37 | 2.5 |
| Standard error due to natural variability | 0.0061 | 0.0072 | 0.0017 | 0.145 | 0.13 |
| Is error smaller than 2 standard errors? | No | Yes | Yes | Yes | Yes |
| error relative to simulated $\mu$ | -23% | -8.5% | 0.9% | 49% | -4.0% |

## 3.3 Constrained quadratic optimization

With the linear model of climate responses to GHG and SAI forcing, we can explore the possible climate responses to different combinations of SAI strategies within the chosen design space. Most importantly, we can address questions such as:

1. Which climate goals can be simultaneously managed?

2. For different choices of climate goals can we find a "better" strategy than any of the existing strategies?

3. How do those strategies depend on the choice of goals?

We only consider strategies that reduce the global mean temperature to the reference level (or equivalently, only consider metrics defined relative to a reference world without SAI but with the same global mean temperature), and we only consider quadratic (and hence convex) optimization functions. Neither of these assumptions is strictly necessary; they are chosen for simplicity.

The design space consists of all possible linear combinations of the seven SAI strategies noted in Table 1, or equivalently any set of injection rates across the 7 latitudes, and in either case subject to a constraint that injection rates at all latitudes are non-





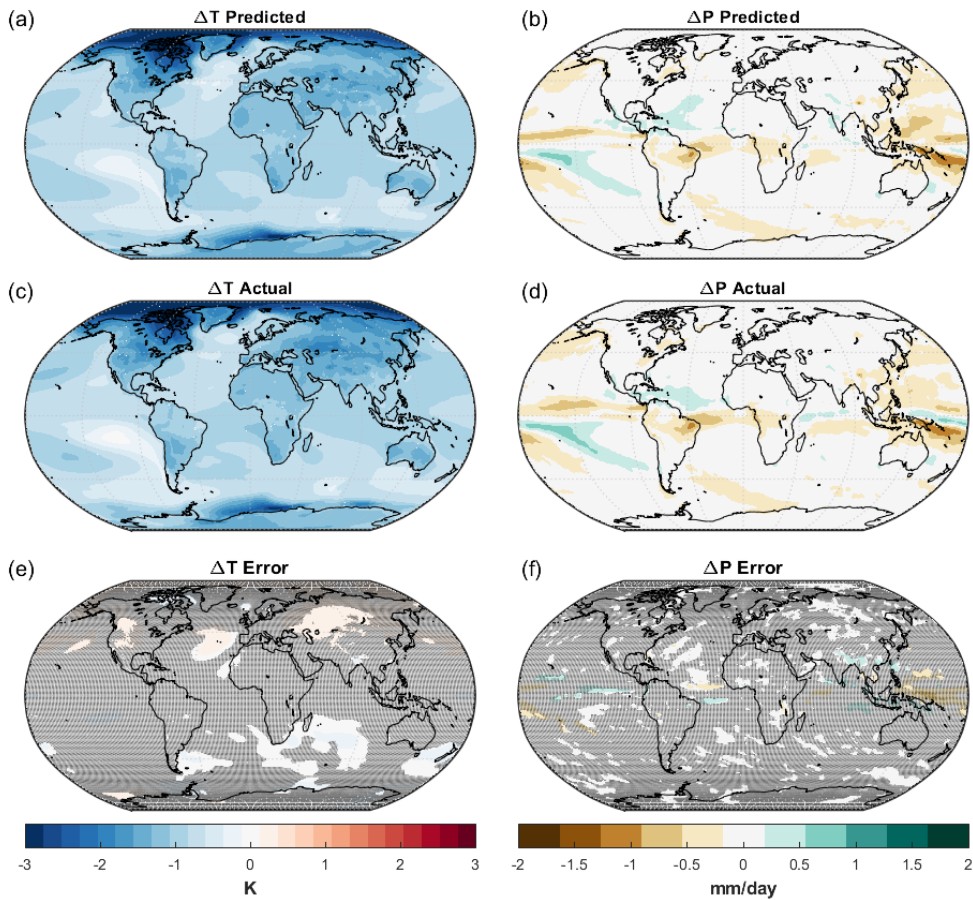

**Figure 2.** Comparison of predicted and simulated regional response of temperature and precipitation to 1°C global cooling provided by the multi-objective strategy. Changes in regional temperature and precipitation predicted by the linear model are shown in (a) and (b), respectively. Changes in regional temperature and precipitation simulated in CESM2(WACCM6-MA) are shown in (c) and (d), respectively. The differences between the predicted and simulated values are presented in (e) and (f); shaded areas indicate where the difference is smaller than two standard errors of natural variability.





**Table 4.** Injection rates required in these seven strategies to provide $1°$ C global mean surface cooling, and the corresponding decision variables (fraction of the total global mean cooling provided by each of the strategies) in the optimization problem. Injection rates in this table are averaged over 2050-2069.

| SAI strategy | Total injection rate at all latitudes per $1°$ C cooling [Tg/yr] | Injection rate at each latitude [Tg/yr] | Fraction of the total global cooling |
|---|---|---|---|
| 0N | $\delta_1 = 17.4$ | $\delta_1$ | $x_1$ |
| 15N+15S | $\delta_2 = 12.6$ | $\delta_2^{15N} = \delta_2^{15S} = 0.5\delta_2$ | $x_2$ |
| 30N+30S | $\delta_3 = 10.9$ | $\delta_3^{30N} = \delta_3^{30S} = 0.5\delta_3$ | $x_3$ |
| 60N+60S | $\delta_4 = 16.4$ | $\delta_4^{60N} = \delta_4^{60S} = 0.5\delta_4$ | $x_4$ |
| 60N | $\delta_5 = 16.8$ | $\delta_5$ | $x_5$ |
| Multi-objective | $\delta_6 = 11.8$ | $\delta_6^{30N} = 0.04\delta_6$ <br> $\delta_6^{15N} = 0.30\delta_6$ <br> $\delta_6^{15S} = 0.52\delta_6$ <br> $\delta_6^{30S} = 0.14\delta_6$ | $x_6$ |
| 30N | $\delta_7 = 12.3$ | $\delta_7$ | $x_7$ |

negative. We choose to optimize over $x_k$, $k = 1 \ldots 7$, the fractional cooling provided by each strategy; the relationship between these variables and the injection rates at each latitude are shown in Table 4 and are used both to enforce the non-negativity constraint and to express conclusions in terms of injection rates.

The objective function is the weighted mean square of the differences between the predicted values under SAI, and the corresponding reference or target values of all considered climate variables:

$$\underset{\boldsymbol{x}}{\arg\min} \quad J(\boldsymbol{x}) = \sum_{i=1}^{i=m} w_i \left( \sum_{k=1}^{k=7} x_k \beta_{k,i} \right)^2 \tag{9}$$

where $\beta_{k,i}$ denotes the change in the $i$th climate variable per $1°$ C of global cooling in response to the sum of GHG forcing from SSP2-4.5 and SAI forcing from the $k$th strategy, and $w_i \geq 0$ denotes the weight of the $i$th climate variable. The solution to this

constrained quadratic optimization problem is found using the interior-point method, with the MATLAB function "quadprog." After solving for $x_k$, we can solve for the corresponding injection rates at each of the seven latitudes.

## 4   Managing large-scale climate variables simultaneously

Existing studies have only explored strategies that can manage up to three climate goals at the same time, by adjusting injection rates at up to 4 latitudes simultaneously (but only adjusting three degrees of freedom independently; see MacMartin et al.

(2017)). In principle, with seven independent degrees of freedom available, it might be possible to simultaneously satisfy seven distinct goals. However, this is not in general possible. Using the constrained quadratic optimization formulation introduced above, we can evaluate whether there exists a strategy that can simultaneously achieve $m$ climate goals ($m \geq 1$) by looking for




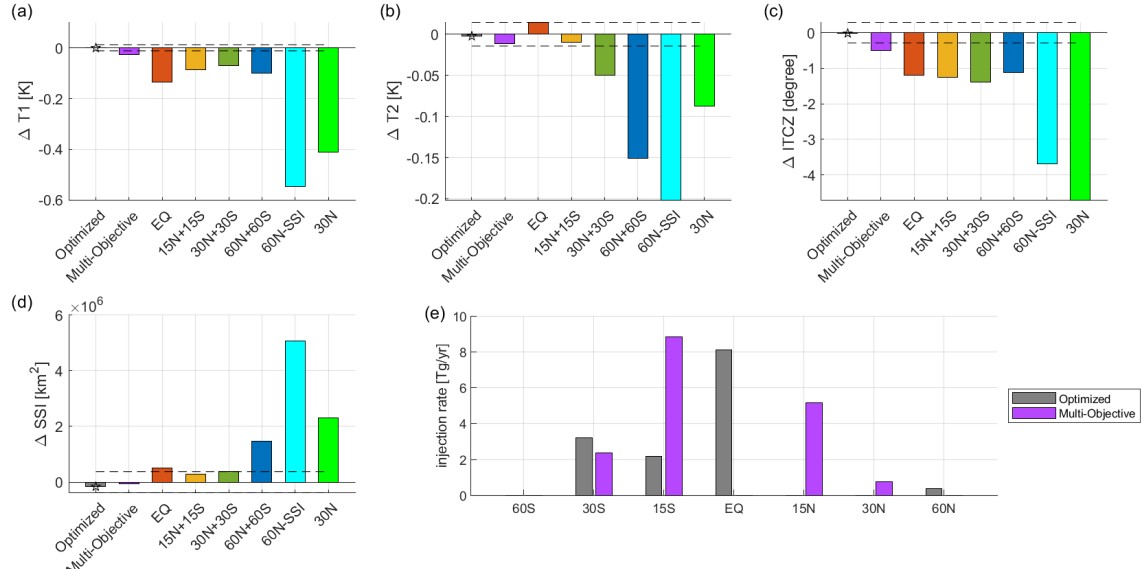

**Figure 3.** The value of $T_1$, $T_2$, ITCZ and SSI (panels a-d, respectively) relative to the reference period (2008–2027) under the optimal strategy and the seven existing strategies, which all maintain $T_0$ at the reference level. The dashed lines represent $\pm 2$ standard errors of the 20-year averages of the reference period due to natural variability. Panel e shows the injection rates at different latitudes for the multi-objective and optimized strategies.

solutions $J(x) = 0$ (or close to zero). We consider the following five climate goals, in addition to maintaining the global mean temperature $T_0$: $P_0$, $T_1$, $T_2$, ITCZ, and SSI. This was explored by Lee et al. (2020), but only 30°S, 15°S, 15°N, and 30°N were

used in that study

We first note that while $T_0$ and $P_0$ can indeed be simultaneously managed (the 60N+60S strategy nearly does this, by "undercooling" the tropics where most of the precipitation is), these two cannot be managed at the same time as any of the additional variables above. We therefore restrict our attention to the remaining large-scale metrics and assess whether $T_0$, $T_1$, $T_2$, ITCZ, and SSI can be managed simultaneously. This is not quite achievable in this climate model, but nearly so, with

the residual differences in these metrics relative to the reference period smaller than the uncertainty due to natural variability. Figure 3 compares this injection strategy and the seven existing strategies in terms of changes in $T_1$, $T_2$, ITCZ and SSI relative to the reference period (2008–2027). The injection distribution across latitudes in this optimized strategy is similar to that of the multi-objective strategy, but with most of the 15N+15S injection being replaced by equatorial injection.

## 5   Trade-offs in minimizing regional climate changes

Past strategy design has typically considered targeting several large scale climate metrics (e.g., Kravitz et al., 2017; Lee et al., 2020). However, focusing solely on large-scale climate metrics can potentially be misleading; e.g., the 60N+60S strategy can





simultaneously balance both global mean temperature and global mean precipitation, but there is considerable spatial variation in temperature in particular relative to the reference climate. In this section, we therefore turn our focus to minimizing regional (grid-scale) changes from the reference period, and in particular on trade-offs, through two examples: First, changing the
relative weighting between area-weighted mean-square temperature changes and precipitation changes over the entire Earth surface, and then between temperature and precipitation minus evaporation (P-E), over land only. There are of course an infinite number of cases one could explore, but two is sufficient to illustrate broader conclusions. The objective function representing the area-weighted mean-square temperature and precipitation differences from the reference period is given by:

$$J(\boldsymbol{x}) = \gamma \left( \sum_j \sum_i w_{ij} \left( \sum_{k=1}^{k=7} x_k \beta_{k,T}^{ij} \right)^2 \right) + (1-\gamma) \left( \sum_j \sum_i w_{ij} \left( \sum_{k=1}^{k=7} x_k \beta_{k,P}^{ij} \right)^2 \right) \tag{10}$$

where $i$ is the latitude index, $i \in [1, 192]$, and $j$ is the longitude index, $j \in [1, 288]$. The change in temperature in the region indexed as $ij$ is $\sum_{k=1}^{k=n} x_k \beta_{k,T}^{ij}$ and the change in precipitation in the corresponding region is $\sum_{k=1}^{k=n} x_k \beta_{k,P}^{ij}$, and the weighting is chosen here to weight by area. The values of $\gamma$ and $1-\gamma$ are the relative weights for temperature and precipitation, respectively, where $\gamma \in [0, 1]$.

The optimal solutions as a function of $\gamma$ form a Pareto front, which shows the trade-off between prioritizing tempera-
ture vs precipitation; this is shown in Figure 4 along with the existing strategies. The top-left-most point on the Pareto front is temperature-optimal ($\gamma = 1$), and the bottom-right-most point on the Pareto front is precipitation-optimal ($\gamma = 0$). The hemispherically-balanced strategies are much closer to the Pareto front than the single-hemisphere strategies and non-SAI case. In this model, the r.m.s. temperature change is greatly reduced regardless of the strategy, except for the cases that are not at all hemispherically balanced.

While the predicted behavior of the optimized cases must by definition be "better" (for the metrics being optimized) than any of the existing strategies, the most critical question is how much better, and in particular two slightly related questions (i) whether the differences are meaningful given the information we have, and (ii) if they are meaningfully different, whether the differences are large enough to matter in a deployment. The extent to which general conclusions from a single model study would hold in the real world also matters; of course specific quantitative results will differ, but even with a single model it is
certainly valuable to gain insight into the extent to which multi-latitude optimization is likely to help achieve a desired climate, the extent to which the solution depends on the chosen metric, etc.

There are two sources of error in predicting the response to a linear combination of injection choices in this particular climate model. One is the extent to which the results will inevitably differ from the linear and quasi-static approximation, and the other is that the estimate of the response to any simulated strategy includes the effects of natural variability. Figure 4 (right panel)
illustrates the magnitude of both of these effects relative to the differences across strategies.

First, the purple square represents the value of the Multi-Objective strategy predicted using the linear combination method from section 3.2. The predicted value is about half-way between the simulated multi-objective strategy and the temperature-optimal point on the Pareto front. This suggests that the difference between the multi-objective strategy and the temperature-optimal strategy may not be as significant as the plot suggests.



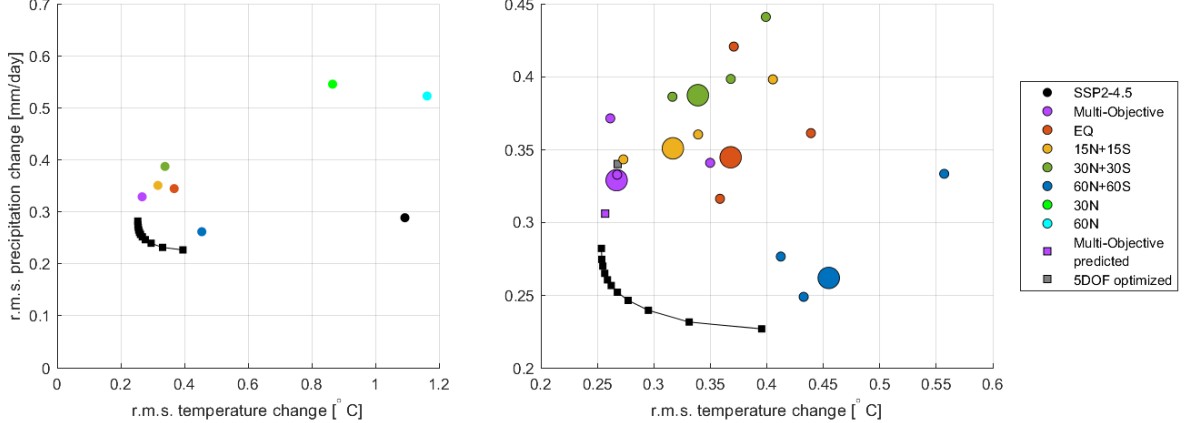

**Figure 4.** A comparison between the r.m.s. temperature changes and r.m.s. precipitation changes in the optimal strategies and the existing strategies. The black squares form the Pareto front. The left panel shows the ensemble average for all strategies. The right panel is a zoomed-in view which shows only the hemispherically-balanced strategies, along with some estimation of the effects of variability and of nonlinearity for context. Large circles represent ensemble-averages, and small circles represent individual ensemble members, giving an estimate of the magnitude of the influence of variability on these metrics. The purple square is the predicted value for the multi-objective case assuming linearity, from Section 3.2. The grey square is the strategy that maintains T0, T1, T2, ITCZ, and SSI from section 4.

To evaluate the effect of natural variability, Figure 4 (right panel) also shows the r.m.s. temperature and precipitation change for each ensemble member (small circles) and the ensemble means (large circles). This shows how distinct each strategy is in the context of internal variability. The difference between the 15N+15S strategy and the Multi-Objective, EQ, and 30N+30S strategies are of similar size to the variation due to internal variability, making these strategies only detectably different from each other for these metrics because we have multiple ensemble members available. The strategy predicted to optimize r.m.s.

temperature differs from the multi-objective case by an amount comparable to our ability to distinguish them given the finite simulations that we have. The magnitude of this second effect could be reduced by running more and more ensemble members, but of course in the real world there is only ever a single ensemble member; even if one simulated many more ensemble members and used those to more precisely define an optimum strategy (and even if it were the optimum strategy in the real world, not just the model), an observer might not be able to tell the difference between the optimum and the multi-objective

strategy unless the level of cooling was much larger than the level used (roughly 1°C cooling) in the simulations used herein. Arguably, this assumes that climate models adequately represent signal-to-noise ratio of responses to perturbations. However, some studies have suggested that models tend to underestimate the predictable component of forced response, in particular on regional scale (Scaife and Smith, 2018; Gillett et al., 2003).

    Nonetheless, the predicted optima deviate further from the existing strategies as the relative weighting on precipitation is

increased. Figure 5 shows the spatial temperature and precipitation changes from the reference period at different points along the Pareto front. The optimal temperature and balanced strategies have similar patterns, whereas the optimal precipitation





strategy undercools the tropics and overcools the poles, which is consistent with results of prior studies that show that SAI at higher latitudes has a lesser effect on precipitation (Zhang et al., 2024).

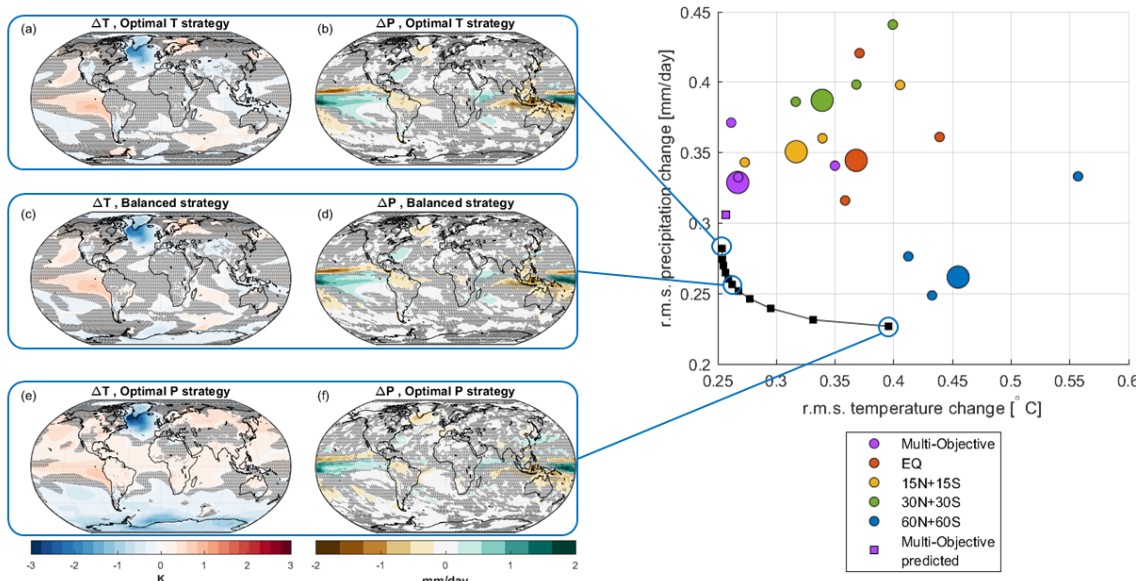

**Figure 5.** Maps on the left show the deviation in temperature and precipitation from the reference period for 3 different optimal temperature vs precipitation strategies. The scatter plot on the right shows which points on the Pareto front these strategies refer to, as well as the r.m.s. temperature and precipitation changes in existing strategies. Large circles are ensemble averages, and small circles are individual ensemble members, as in Figure 4.

There are two broad observations from exploring the trade-off between minimizing the spatial mean-square temperature
deviations from a reference climate, and the spatial mean-square precipitation deviations. First is that the existing multi-objective strategy is pretty good at minimizing the overall mean-square temperature changes. The 30N+30S and 15N+15S hemispherically-symmetric strategies might be similarly good in other models that do not have a strong cloud fast-adjustment to $CO_2$ that is a key factor responsible for CESM2(WACCM)'s need to inject more in the southern hemisphere to balance T1 (Fasullo and Richter, 2023). Second is that it seems likely that the mean-square precipitation response could be improved
somewhat relative to the existing multi-objective strategy without substantially affecting the mean-square temperature response, but these differences will likely be too small to matter in the early decades of deployment when the cooling levels are small. The simulations on which the results herein are based cooled by of order 1°C, yet the scatter from natural variability alone is substantial relative to the gains from optimization. Any realistic deployment scenario is likely to take at least decades, if ever, to reach such a substantial cooling level.





Figure 6 shows the SO$_2$ injection rates across the seven latitudes for the strategies along the Pareto front discussed above. Temperature-oriented strategies have most or all of the injection at the equator, 60°N, and 60°S, whereas precipitation-oriented strategies distribute injection across more latitudes. Notably, equatorial, 60°N, and 60°S are favored by this optimization algorithm despite being the latitudes that are not part of the multi-objective strategy, which is the best-performing pre-existing strategy based on the rms temperature metric.

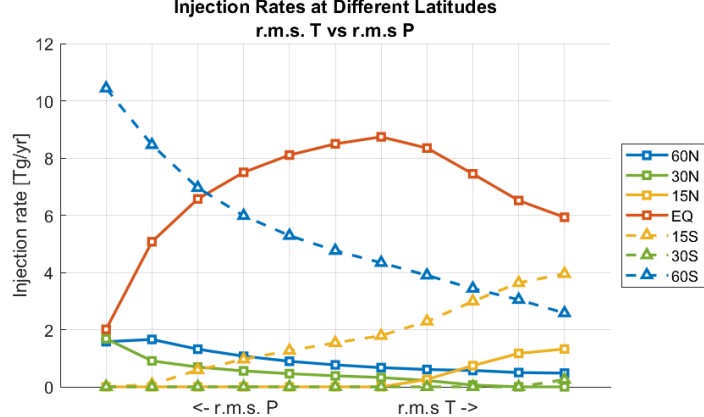

**Figure 6.** SO$_2$ injection rates at each latitude throughout the Pareto front for r.m.s. temperature vs precipitation

.

There are of course an infinite set of metrics that could be considered, and temperature and precipitation are just two; we therefore consider a second example to see if similar conclusions hold. While temperature and precipitation are commonly explored, precipitation minus evaporation (P-E) may be more relevant to impacts, and further, it is changes over land that are most relevant for most impacts. Figure 7 shows the results of optimizing for temperature over land vs P – E over land. The error induced by the linear approximation method is larger than the difference between the multi-objective strategy and much

of the Pareto front. Thus not only is the difference too small to be meaningful between the existing multi-objective strategy and the strategy predicted to minimize land-average temperature, but this holds for a considerable portion of the Pareto front.





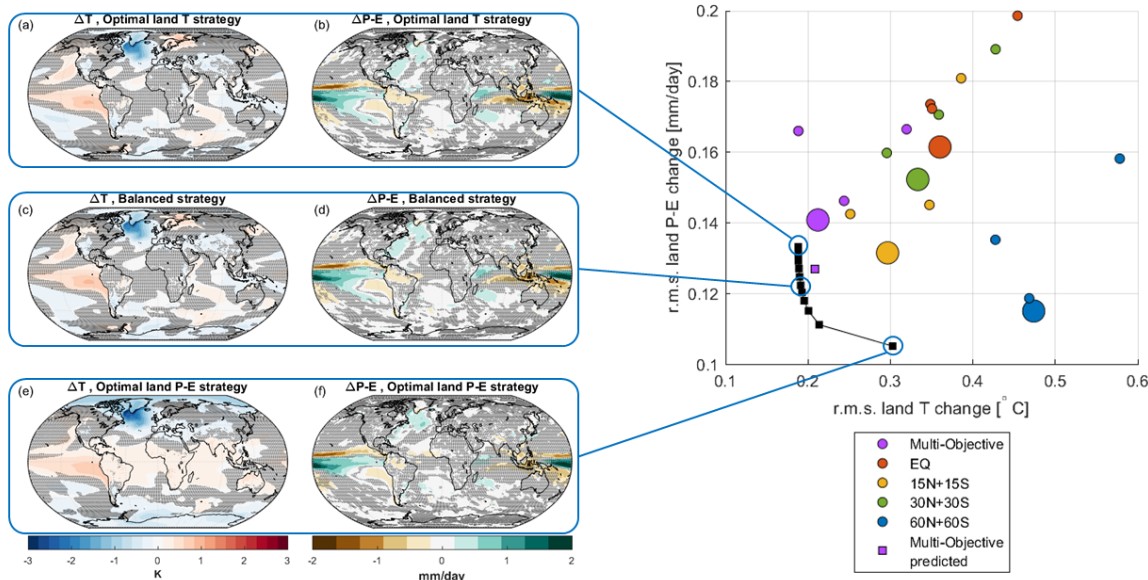

**Figure 7.** Same as figure 5, but for r.m.s. land temperature vs r.m.s. land precipitation minus evaporation

Figure 8 shows the injection rates for these strategies. These land-oriented strategies utilize more latitudes than globally-oriented ones.

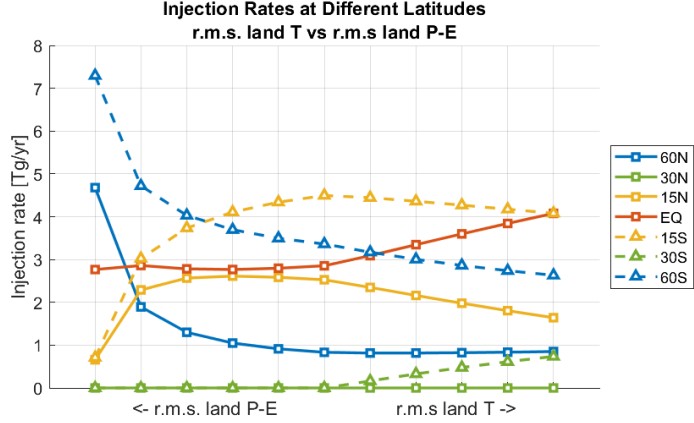

**Figure 8.** SO$_2$ injection rates at each latitude throughout the Pareto front for r.m.s. land-only Temperature versus land-only P-E.

The analyses for the r.m.s. temperature vs precipitation trade-off and the r.m.s. land temperature vs r.m.s. land precipitation
minus evaporation trade-off were repeated with the values in each grid cell normalized by the standard deviation in that grid cell during the reference period. The results were very similar and can be found in the supplemental material.





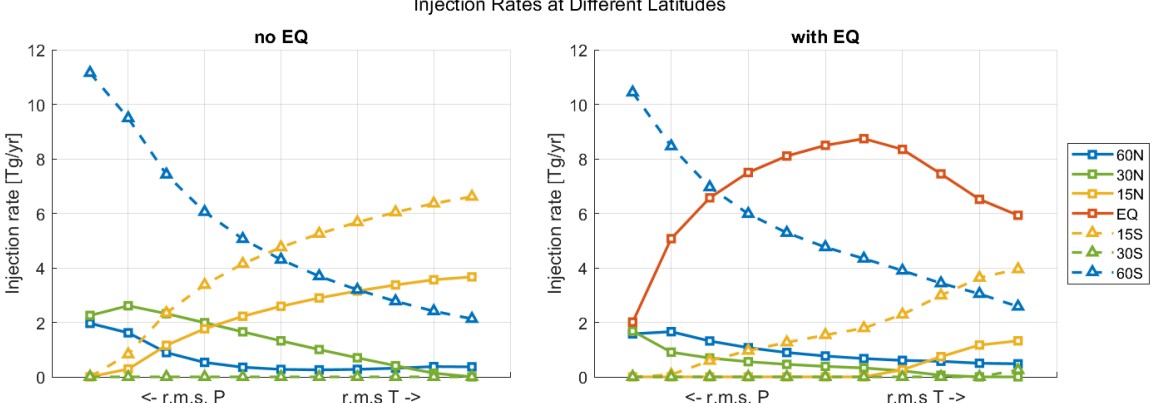

**Figure 9.** SO$_2$ injection rates at each latitude throughout the Pareto front for r.m.s. temperature vs precipitation with the equator removed as an option (left) and with all 7 latitudes available (right)

The optimization framework also allows straightforward exploration of additional questions, such as how important it is to be able to inject at specific latitudes. For example, the optimized strategies rely heavily on equatorial injection, yet the multi-objective strategy does not include equatorial injection at all. In order to investigate how much the injection at the equator contributes to the optimality of the found optimal strategies shown in Fig. 4, we ran the same optimization with equatorial injection removed as an option. Figure 9 (left panel) shows the injection rates at different latitudes for this case compared to the full 7-latitude case. The equatorial injection is mostly replaced with 15°N and 15°S. Figure 10 shows the pareto front for the no-EQ case compared to the full 7-latitude case. The no-EQ pareto front has slightly larger residual climate changes, but the difference is far less than the error potentially induced by the linear quasi-static approximation used. This shows that injection strategies utilizing fewer number of injection latitudes can produce indistinguishably different responses, suggesting that there may be effectively fewer than the 6-8 degrees of freedom found by Zhang et al. (2022).




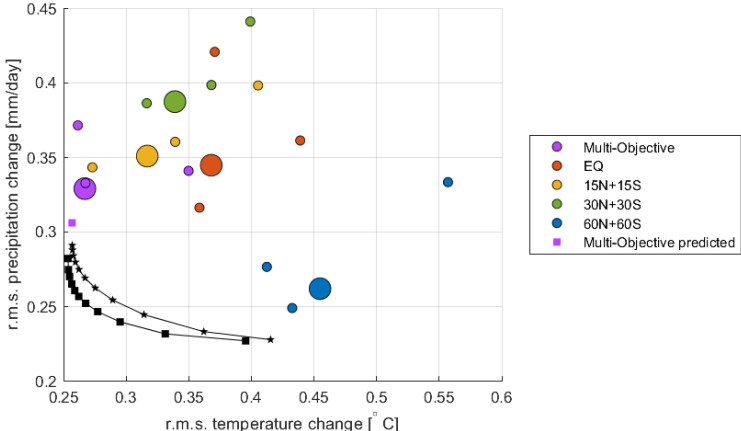

**Figure 10.** Same as figure 4, but with the addition of the pareto front of optimizing r.m.s. temperature vs r.m.s. precipitation change with equatorial injection removed as an option (black stars)

## 6  Discussion

Previous work has considered designing SAI strategies by selecting at most three distinct climate goals, selecting at least as many latitudes of injection, and then designing strategies to balance those climate goals by adjusting injection rates across the

different latitudes. However, research suggests that there should be of order 6–8 meaningfully independent degrees of freedom available by selecting additional injection latitudes and/or seasons. This raises the obvious question of the extent to which that would allow "better" SAI strategies to be designed, where better is of course only meaningful in the context of specifically chosen climate goals, and thus it is also critical to know how much the optimal strategy depends on the choice of goals (e.g., on who gets to decide what matters and what doesn't).

This study introduces a method of creating SAI strategies that are optimized to meet certain climate goals. With 6–8 degrees of freedom, it should be possible to meet more than 3 climate goals simultaneously, and indeed we have demonstrated that this is true; e.g., it is possible to simultaneously maintain global mean temperature, interhemispheric and equator-to-pole temperature gradients, as well as the latitude of ITCZ, and Arctic September sea ice extent.

However, there are many more variables that matter than available degrees of freedom, and it is not possible to fully restore

the entire climate state back to an earlier reference period. (This is not entirely due to inherent characteristics of SAI, but also partially a result of the fact that the earlier reference climate is in a transient, and not equilibrated state (Duffey and Irvine, 2024). We explore this through several examples. First, we consider how well regional temperature and precipitation goals can be met simultaneously. There is an inherent trade-off between keeping the spatial r.m.s. of temperature and the spatial r.m.s. of precipitation as close as possible to reference period levels – better performance for one will result in worse performance for

the other, leading to a Pareto-optimal front as the relative weighting is changed. For any such relative weighting of temperature and precipitation, there is a strategy that meets these goals better than any strategy that has been simulated thus far. However,





the extent to which these strategies are meaningfully better or detectably different from existing strategies is not so clear. The figures in section 5 show that the difference between the simulated and predicted multi-objective strategy is similar to the difference between the simulated multi-objective strategy and the temperature-focused optimal strategies on the Pareto front. This means that it is not possible to determine, at least in this study, if there is a strategy that can actually do better than the multi-objective at maintaining temperatures around the world, at least using the rms metric. Figure 4 shows that natural variability, at least in the model, is similarly large, adding another layer of uncertainty as to whether such a difference could be detected in models or in the real world. There is, though, a strategy in the center of the Pareto front that maintains temperature just as well as the multi-objective strategy but is likely detectably better at maintaining precipitation for of order 1°C cooling or more – at low cooling, these differences would be more difficult to detect.

Changing the goals to r.m.s. temperature change over land and r.m.s. precipitation minus evaporation change over land yields similar results. The difference between the multi-objective strategy and the strategy that minimizes land-averaged temperature changes is perhaps even less significant in this case. The spatial patterns of the responses are very similar to those from the global temperature vs precipitation optimization, suggesting that the exact choice of metric used may not be particularly important.

One interesting detail of the results is the injection latitudes chosen by the optimization algorithm. Most of the strategies focus the injection at the equator, 60°N, and 60°S. When designing the multi-objective strategy (MacMartin et al., 2017; Kravitz et al., 2017), the equator was not chosen as an injection latitude because it results in a high aerosol concentration in a narrow latitude band near the equator, while dividing that injection between 15°N, and 15°S produces a more uniform AOD. The high latitudes 60°N and 60°S were also not originally chosen as the aerosol lifetime in the stratosphere is much shorter compared to injection at lower latitudes (MacMartin et al., 2017). Despite this, their combined effect does the best job at counteracting the effects of GHG on temperature and precipitation. However, removing equatorial injection as an option for the optimization results in that injection being mostly shifted to 15°N, and 15°S, and the ensuing climate responses are not detectably worse. This suggests that there are fewer meaningfully-distinct degrees of freedom than the 6-8 suggested by Zhang et al. (2022). We note that the underlying assumption here is that climate models adequately represent signal-to-noise ratio of responses to perturbations. However, some studies suggest that even state-of-the art climate models can underestimate the predictable component of forced response, in particular on regional scale (Scaife and Smith, 2018; Gillett et al. 2003), which if true could influence the results here.

A key question is whether there are additional strategies that ought to be considered in understanding the range of climate responses from SAI beyond those already identified – for example, are there ways that it could be optimized to reduce side effects beyond those strategies already considered in the literature. A critical observation in answering this question is that the amount of cooling that is likely in the first few decades of deployment is probably much less than 1°C, and by that time there will be substantially more information available about the effects of SAI and its dependence on strategy. For this range of cooling (much of the analysis herein is scaled per degree cooling), our conclusion is that the differences between existing strategies and optimized strategies are often not very large, particularly for temperature-based metrics. There may be strategies



that do a little bit better on precipitation-based metrics, but even there the differences may not be large enough to change the answer of whether or not to begin a deployment of SAI.

There are several ways in which this study could be built upon in the future. The first is to use more climate models; this study only uses one: CESM2-WACCM(MA), and performing this analysis in other models may yield different results (Visioni et al., 2023a; Henry et al., 2023; Wells et al., 2024). However, it is unlikely that the qualitative conclusions would change (Kravitz et al., 2014). Even with just one climate model, more ensemble members of each basis strategy could be run. This would result in greater accuracy of the results as well as increased power in detecting differences between strategies. It would not, however, change the conclusion that many of the differences found are comparable or smaller than internal variability.

Another way to improve this study would be with a better estimation method. This study uses a linear quasi-static approximation, the limitations of which are discussed in section 3.2. Development of dynamic climate emulators for SAI has been gaining attention recently (Farley et al., 2024; Beckage et al., 2023). If a more advanced emulator that is nonlinear or accounts for the different time-scales that exist in the climate system were developed, it could be used to perform this analysis with more accurate results.

This study only uses a small set of objective functions for optimization, and the analysis could be extended to consider other metrics, including other climate variables and non-annual-mean quantities. Herein we weight departures from a reference climate quadratically, but other dependencies could be explored, including non-symmetric ones.

Lastly, exploring the space of what can be achieved with SAI is valuable, but actually designing a strategy to achieve these goals would require additional effort due to the inevitable uncertainties and nonlinearities; even achieving something close to a predicted optimum in the same climate model would likely require use of a feedback controller that adjusts the injection at each latitude over time.

*Author contributions.* EB and YZ conducted all analyses and wrote the paper, with editing from DGM, EMB, DV, and BK. YZ and DGM conceived the study, with input from all authors. EMB and DV assisted with conducting simulations.

*Competing interests.* At least one of the (co-)authors is a member of the editorial board of Earth System Dynamics. The authors have no other competing interests to declare.

*Acknowledgements.* The authors would like to acknowledge high-performance computing support from Cheyenne (https://doi.org/10.5065/D6RX99HX) provided by NCAR's Computational and Information Systems Laboratory, sponsored by the National Science Foundation. Support for E. Brody, Y. Zhang, and D. G. MacMartin was provided by the National Science Foundation through agreement CBET-2038246. Support for D. Visioni was provided by the Cornell Atkinson Center for a Sustainable Future. Support for B. Kravitz has been provided by the National Science Foundation (grant no. SES-1754740), NOAA's Climate Program Office, the Earth Radiation Budget initiative (grant no. NA22OAR4310479), and the Indiana University Environmental Resilience Institute. Support for E. M. Bednarz has been provided by the





National Oceanic and Atmospheric Administration (NOAA) cooperative agreement NA22OAR4320151 and the Earth Radiative Budget (ERB) program. The CESM project is supported primarily by the National Science Foundation.



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
