# Peer review of "Using Optimization Tools to Explore Stratospheric Aerosol Injection Strategies"

_EGUsphere, 2024_

## Author Comment (AC1)

**Response to Reviewer #1**

We thank the reviewer for taking the time to review our manuscript, and for their supportive comments.

Note that in between the time of submission and now, two more ensemble members of the 60N constant injection simulation became available, so we redid the analysis with these instead of the Arctic High simulation. This did not change any of the conclusions, but we now have 3 ensemble members of each of the seven strategies used in this study. Figures and results have been updated accordingly.

Brody et al. are interested in selecting latitudes at which to emit aerosol precursor gas into the stratosphere in order to optimize climate change with regard to some scalar metrics such as temperature or precipitation changes. They do so on the basis of available climate model simulations and apply a simple emulator in the optimisation process.

The study in general is well conducted and the manuscript is well written.

As the only major suggestion, it would be very useful to test the results by implementing the optimal choice of injection into a different Earth system model (see also lines 142-143 that somewhat alludes to this idea).

We agree that simulating one or more of the optimal injection strategies in another climate model would be beneficial. However, we do not have access to other climate models for this project, so this will have to remain outside of the scope of this study. We leave similar analysis in other climate models to future studies, as suggested in the discussion section, lines 398-401.

Else I propose some minor edits.

l95 – It would be useful to motivate this surprising use of another, very differently conducted, simulation at this point.

While the "Arctic High" simulation from Lee et al adjusted the injection rate to maintain a sea ice target, the actual injection resulting injection rates were not too dissimilar from a constant 12 Tg/yr; however as we now have access to additional ensemble members of fixed injection for 60N, we now avoid this extra complication and we now use only the 3 ensemble members of the fixed-injection 60N injection simulation.

l114-115 – to this point, the reader assumed the assumption of linearity is "validated" (e.g. abstract, line 11). If it turns out very problematic, this needs to be explained instead of assuring the reader about validation.

We have provided a detailed linear approximation evaluation in Section 3.2 to show that the errors due to linear approximation are generally small compared to internal variability. Of course, the actual responses of the climate system are not linear and linearity is only an approximation; the question is whether the errors introduced from making this approximation are large or not (relative to the differences between existing and optimal strategies, as well as internal variability, for example). If the linear assumption turned out to be an unacceptably bad one, we wouldn't have written this paper.

l121 / Table 1 – acronym "SSI" needs to be explained

Fixed.

l146 / Eq. 1 – a noise/natural variability term is missing

The equation refers only to the forced response, therefore we did not include a natural variability term.

l152 – what quantity related to ITCZ, its annual-mean zonal-mean latitude?

Yes, we are referring to the annual-mean zonal-mean ITCZ. We will clarify the zonal-mean part; annual mean is already stated at the end of the paragraph.

l155 – or higher moments of the quantities, especially extremes would be interesting.

Analyses of the dependency of higher moments, or extreme events, would definitely be of interest, and we will work towards it in future works. However, aside from mentioning this possibility now, we feel like further analyses fall beyond the scope of this work.

l174 – define "adequately" linear

Figure S5 shows the scalar variables used in this study plotted against $T_0$ for SSP2-4.5. This shows that these variables scale approximately linearly with respect to $T_0$, and errors due to nonlinearities are smaller than the noise. Note that for SSI, the linear approximation is only valid for the first few decades, before it approaches zero. This is why 2030-2049 was chosen instead of 2050-2069 as the future time period for SSI.

This will be added to the supplementary material.

[Figure]

Figure S5: Select variables vs global-mean surface temperature ($T_0$) for the SSP2-4.5 simulations from 2000-2069. Solid black lines are ensemble means, dashed black lines are individual ensemble members, and red lines are a linear fit with the slope equal to the α values from Table 2.

l179 / Eq. 2 – a noise term is missing

The equation refers only to the forced response.

l184 / Table 2 – "values". Also I gather \alpha is for the SSP2-4.5 column, and all other columns list \mu? This should better be explained in the caption. The signs in the ITCZ location require explanation.

We will make these clarifications in the caption.

l192 – define T_0 (=T_0,warming)

We will make this fix.

l202 – How is this possible (in terms of seeding), a negative coefficient (see also line 222)? And also, why such a complicated combination, instead of straightforward combination?

A coefficient of -5% is possible for the 30N simulation because of the +18% coefficient for the 30N+30S simulation, resulting in a positive amount of injection at 30N. (Note that we accidentally switched the injection rates between the NH and SH in the manuscript. The SH should have more injection in the Multi-Objective strategy. This will be fixed) The same reasoning holds for 15N. The reason that such a complicated combination was chosen is that the Multi-objective run ramps up the injection rate over time, so we wanted to use simulations

that also ramp up their injection rates (15N+15S, 30N+30S) as much as possible over simulations with constant injection rates (15N, 15S, 30N, 30S)

l209 /Table 3 – additional digits required for P_0 and SSI.

Number of decimal places for these values were chosen based on the standard error in a consistent manner for each variable.

---

## Author Comment (AC2)

**Response to Reviewer #2**

We thank the reviewer for taking the time to review our manuscript, and for their supportive comments.

Note that in between the time of submission and now, two more ensemble members of the 60N constant injection simulation became available, so we redid the analysis with these instead of the Arctic High simulation. This did not change any of the conclusions, but we now have 3 ensemble members of each of the seven strategies used in this study. Figures and results have been updated accordingly.

Brody and coauthors provide a novel and useful analysis of design space of stratospheric aerosol injections in CESM2-WACCM. Specifically, it is interesting to see the trade-offs between optimizing temperature and precipitation responses extends to the pattern of response when designing SAI interventions, in addition to the well known trade offs in the global mean responses. The manuscript is well written and is largely ready for publication. However, I believe the analysis would be benefit substantially from a more rigorous assessment of the uncertainty due to natural variability. I am concerned that internal variability presents a significant constraint on the ability to optimize climate responses that must be quantified. Particularly when computing the sum of anomalies, the noise in the pattern can increase substantially as the number of signals being combined is increased (i.e., the sum of variances than the mean of variances). One approach may be to compute the Pareto front with all 18 possible combinations of individual ensemble members from each of the existing strategies. This could be used to generate a "cloud" of such Pareto fronts and uncertainty ranges for the dots in Fig. 6 and 8.

Good idea! Showing the pareto fronts generated by individual ensemble members is the sensible complement to showing the individual ensemble members of the simulated strategies. Note that the background emissions in the three individual ensemble members of SSP2-4.5 simulations are not exactly the same, so SAI simulations corresponding to different SSP2-4.5 ensemble members do not share the same background emissions and are not directly comparable. However, we have compared the pareto fronts generated by the individual ensemble members in the same ensemble (i.e., one for the 1st ensemble, one for the 2nd ensemble, and one for the 3rd ensemble), and concluded that there always exist better strategies than the existing ones.

The right plot in Figure 1 is what it looks like with the pareto front generated with each of the ensemble individually, shown as black triangles. As you can see, the spread is rather large compared to the distance between the pareto front and existing strategies, suggesting that the results of optimization may not be significant.

[Figure]

Figure 1. Pareto Front of ensemble means (left) and Pareto Front of individual ensembles (shown as black triangles) (right).

Figure 2 is what it looks like with each ensemble member of the existing strategies and pareto front shown on its own panel. The locations of simulated strategies and the pareto front are different in each panel, but the pareto front is closer to the origin than the existing strategies in each panel. This suggests that although there is a large spread in the resulting climate with each strategy, there are always better strategies than the existing ones.

[Figure]

Figure 2. Pareto Front of the optimized strategies using individual ensembles in the (a) first ensemble, (b) second ensemble, and (c) third ensemble, compared to the simulations of the existing strategies in the same ensemble.

Additional minor comments follow.

- Line 4: "latitudes of injection" -> "latitudes and altitudes of injection"

We specifically don't look look at varying altitudes, as it doesn't significantly change the resulting climate pattern, only the amount of injection needed to achieve that pattern (Zhang et al. 2022)

- Line 5: ", managing up to" -> ". For example, managing up to"

We will make this change.

- Line 113:" the Atlantic Meridional Overturning Circulation" -> "rate dependent responses, such as the Atlantic Meridional Overturning Circulation (Hankel 2024)". Hankel, C. 2024 https://www.pnas.org/doi/abs/10.1073/pnas.2411357121

We will make this change.

- Line 205-206: How is the standard error computed? Is is computed using variance across an initial condition ensemble?

Standard error is computed using annual-mean values from each of the 20 years from 2008-2027 in each of the 3 ensemble members of the SSP2-4.5 (without SAI) simulation. Autocorrelation correction is applied to account for the fact that consecutive years are not fully independent.

- Line 370: "at low cooling" -> "at lower cooling"

We will make this change.

- Line 403: "greater accuracy" -> "greater precision"

We will make this change.

- Fig 6,8,9: I assume since the 1C cooling is the reference point, the overall emission magnitude can change? It would be useful to see total emissions would help see if there are overall SAI efficiency differences between the optimizations

This is a good idea. I'll add a "total" line to these plots.